# Molecular Dynamics Simulation Study of Aluminum–Copper Alloys’ Anisotropy under Different Loading Conditions and Different Crystal Orientations

**DOI:** 10.3390/ma17164162

**Published:** 2024-08-22

**Authors:** Xiaodong Wu, Wenkang Zhang

**Affiliations:** 1School of Information Science & Engineering, Changsha Normal University, Changsha 410100, China; 2State Key Laboratory of Advanced Design and Manufacturing Technology for Vehicle, Hunan University, Changsha 410082, China; 3College of Engineering, Anhui Agricultural University, Hefei 230036, China

**Keywords:** aluminum–copper alloy, crystal orientation, anisotropy, molecular dynamics, deformation mechanism

## Abstract

The commonly used aluminum–copper alloys in industry are mainly rolled plates and extruded or drawn bars. The aluminum–copper alloys’ anisotropy generated in the manufacturing process is unfavorable for subsequent applications. Its underlying mechanism shall be interpreted from a microscopic perspective. This paper conducted the loading simulation on Al–4%Cu alloy crystals at the microscopic scale with molecular dynamics technology. Uniaxial tension and compression loading were carried out along three orientations: X-<$\bar{1}12$>, Y-<$1\bar{1}1$>, and Z-<110>. It analyzes the micro-mechanisms that affect the performance changes of aluminum–copper alloys through the combination of stress–strain curves and different organizational analysis approaches. As shown by the results, the elastic modulus and yield strength are the highest under tension along the <$1\bar{1}1$> direction. Such is the case for the reasons below: The close-packed plane of atoms ensures large atomic binding forces. In addition, the Stair-rod dislocation forms a Lomer–Cottrell dislocation lock, which has a strengthening effect on the material. The elastic modulus and yield strength are the smallest under tension along the <110> direction, and the periodic arrangement of HCP atom stacking faults serves as the main deformation mechanism. This is because the atomic arrangement on the <110> plane is relatively loose, which tends to cause atomic misalignment. When compressed in different directions, the plastic deformation mechanism is mainly dominated by dislocations and stacking faults. When compressed along the <110> direction, it has a relatively high dislocation density and the maximum yield strength. That should be attributed to the facts below. As the atomic arrangement of the <110> plane itself was not dense originally, compression loading would cause an increasingly tighter arrangement. In such a case, the stress could only be released through dislocations. This research aims to provide a reference for optimizing the processing technology and preparation methods of aluminum–copper alloy materials.

## 1. Introduction

By virtue of their high specific strength, great conductivity, and sound thermal conductivity, aluminum–copper alloys are widely used in fields such as automotive manufacturing, aerospace, and the defense industry [1,2,3]. The commonly used aluminum–copper alloys in industry are mainly rolled plates and extruded or drawn bars. The aluminum–copper alloys’ anisotropy generated in the manufacturing process is unfavorable for subsequent applications [4,5]. The formation of anisotropy phenomena cannot be fully explained from a macroscopic perspective. It is, therefore, necessary to study the atomic structure changes, dislocation distribution changes, evolution mechanisms, and deformation mechanisms that occur during tension and compression loading from a microscopic perspective so as to reveal the underlying mechanism of the mechanical properties of aluminum–copper alloys under the action of tension and compression loading.

In recent years, scholars have conducted extensive research on the anisotropy of the mechanical properties of aluminum alloys. For example, Otani et al. [6] analyzed the cause of the anisotropic tensile ductility of Al–Si alloy manufactured by laser powder bed fusion from a microstructure perspective through the combination of digital image correlation (DIC) strain analysis and in situ SEM observations of tension tests. Choi et al. [7] conducted experimental research on the anisotropy of the mechanical properties of 7075-T6 aluminum alloy sheets. Rahmaan et al. [8] studied the mechanical anisotropy of six series and seven series aluminum alloys through uniaxial tension, simple shear, and through-thickness compression experiments and raised a constitutive model to characterize their anisotropy.

Numerical simulation is a great supplement to traditional theories and experimental methods. Molecular dynamics simulation is a method based on classical Newtonian mechanics to simulate the motion of atoms or molecules, which is conducive to revealing the mechanical behavior and deformation mechanisms of materials at the atomic scale [9].

Currently, some scholars have begun to study the deformation mechanisms and potential mechanisms of material anisotropy with molecular dynamics methods. For instances: Tian et al. [10] studied the mechanical response of nano porous aluminum under impact in different crystal directions through molecular dynamics simulation and found that the collapse rate of holes and the dislocation activity on the surface of holes rely strongly on the impact direction. When the impact is weak, the gap collapses with the fastest speed if the impact is imposed along the <100> direction; when the impact is strong, the cavity collapses with the fastest speed along the <110> direction. An et al. [11] studied the influences of crystal orientation and loading conditions on the mechanical behavior of nanocrystalline titanium through molecular dynamics simulations. Under tension perpendicular to the basic surface, grain redirection and deformation twinning are the main deformation mechanisms. Under compression perpendicular to the basal plane, the HCP-FCC phase transition serves as the deformation mechanism. When the load is perpendicular to the prism plane, the HCP-BCC-FCC phase transition serves as the deformation mechanism for both tensile deformation and compressive deformation. Hou et al. [12] constructed a NT-Cu model with <111> texture through molecular dynamics methods and studied the anisotropic response mechanism and micro deformation mechanism of NT-Cu under dynamic impact. It turned out that when the impact velocity is low, the anisotropy is significant. In addition, there was a noticeable secondary yield phenomenon when loading in the 90° direction. When loading impacts along the 0° and 45° directions, a large number of intrinsic stacking faults emerge in the microstructure; when loaded along the 90° direction, the generated intrinsic stacking faults are relatively few.

Although there are abundant simulation research results about material deformation and phase transformation based on molecular dynamics, the anisotropic deformation mechanism of aluminum–copper alloy, an important industrial material, is yet to be clarified. Due to the single FCC structure of the aluminum–copper alloy, there are rich deformation mechanisms inside it [13]. This paper applied molecular dynamics technology to establish a single crystal model of aluminum copper alloy, simulated the loading of aluminum copper alloy at the microscopic scale, and conducted uniaxial tension and compression loading along three orientations. It analyzed the stress–strain curve, atomic structure changes, dislocation distribution changes, evolution mechanisms, as well as deformation mechanisms, and explored the micro mechanisms that affect the performance changes of aluminum copper alloys, with the aim of providing a reference for optimizing the processing technology and preparation methods of aluminum copper alloys.

## 2. Method and Modeling

In order to compare the changes of aluminum–copper alloys loaded along different orientations, this paper constructed a single crystal model of Al–4%Cu alloy, as shown in Figure 1. The dimension of the model is 178.57 Å × 175.37 Å × 171.83 Å, and the total number of atoms in the model system is 322,124. The corresponding crystal orientation of the model is: X-<$\bar{1}12$>, Y-<$1\bar{1}1$>, Z-<110>.

A fixed layer with a certain atomic thickness was set on the upper and lower sides of the model. The atoms in the fixed layer did not get involved in the deformation process but served as the loading end to simulate the actual loading conditions. During the simulation process, the loading direction was set along the three crystal orientations of the model. Periodic boundary conditions were applied in all three directions, with tension and compression loading applied in each direction, respectively. The embedded atom method (EAM), developed by Liu et al. [14], was used in the simulation process. As such a potential function fits various properties of aluminum–copper alloys well, it has been widely used [15,16]. Before conducting tension and compression loading, the model first relaxed under the NPT (Normal Pressure and Temperature) ensemble in order to balance and eliminate residual stresses in the initial model so as to ensure that the stress in the loading direction starts from 0. The relaxation and loading temperatures were set at 300 K to simulate the loading experiment at room temperature. During the loading process, NVT (Normal Volume and Temperature) ensemble was applied to simulate the Poisson effect of metals under free boundary conditions. The loading rate during simulation was 1 × 10^9^ s^−1^; the step size was 0.001 ps; the final loading strain was 14%. LAMMPS-Stable version (Large-Scale Atomic/Molecular Massively Parallel Simulator which is a molecular dynamics simulation software developed by Sandia National Laboratory (Albuquerque, NM, USA)) software was used for simulation; CAN (Common Neighbor Analysis) algorithm in the visualization software OVITO- version 2.9 (Open Visualization Tool) was applied to identify changes and defects in atomic structure; the DXA (Dislocation Analysis) algorithm was employed for dislocation distribution analysis [17,18,19].

## 3. Results and Discussion

### 3.1. Mechanical Property Analysis

The simulated stress–strain curves of Al–4%Cu alloy single crystals loaded in different directions are shown in Figure 2. In accordance with this figure, it can be seen that the crystal undergoes three stages of elastic deformation, yield, and plastic deformation for both the tension and compression loading, which is consistent with the experimental results previously reported [20]. It can also be clearly seen from the figure that there are significant differences in the stress–strain curves under different directions of loading, whether in tension or compression, which fully demonstrates the anisotropic characteristics of the mechanical behavior of Al–4%Cu alloy under uniaxial loading. The yield strength and yield strain statistics under tension and compression loading in different directions are shown in Table 1.

Figure 2a shows the simulated stress–strain curves for tension in different directions. In accordance with the comparison of the slopes of three stress–strain curves during the elastic deformation stage, the slope is the highest under tension along the <$1\bar{1}1$> direction. This indicates that the elastic modulus is the highest under tension in the <$1\bar{1}1$> direction. The elastic modulus under tension along different directions is ranked from high to low as <$1\bar{1}1$>, <$\bar{1}12$>, and <110>. Under tension along the <$1\bar{1}1$> direction, the yield strength hit its peak, reaching 4.15 GPa. Differences in elastic modulus and yield strength of crystals along different directions are mainly related to the crystal structure. As mentioned by Wang [21], the anisotropy of Ni-based single-crystal high-temperature alloys is mainly due to the crystal structure of alloys with different orientations. On the basis of linear elasticity theory, elastic modulus is a physical quantity that characterizes the strength of atomic bonding. According to Figure 1, it can easily be found that the atomic arrangement on the <$1\bar{1}1$> plane is densest. The tighter the atomic arrangement, the greater the interaction force between atoms, and the more difficult it is to break the metal bond under tension. Therefore, the elastic modulus and yield strength are the highest under tension along the <$1\bar{1}1$> direction, and the yield strain is much greater than that generated under tension along the other two directions.

With the increase in strain, all stress–strain curves under tension show a trend of stress decrease. This is because defects generated in the crystal release stress at this moment, which enters the stage of plastic deformation [22,23,24]. Tension in different directions also shows significant differences during the plastic deformation stage. The stress–strain curve enters a stress plateau stage after yielding under tension along the <$1\bar{1}1$> direction. The stress–strain curve enters a brief stress plateau stage after yielding under tension along the <$\bar{1}12$> direction. After that, as the strain increases, the stress starts to rise, leading to a work-hardening phenomenon. Under tension along this direction, the stress decreases the most in the plastic deformation stage, which shall be attributed to the fact that more dislocations are generated under tension in that direction with the same strain; The yield strain under tension along the <110> direction is the smallest, indicating that plastic deformation is most likely to occur and dislocations are most likely to be initiated. It has to be pointed out that the stress–strain curves under tension along the <$\bar{1}12$> direction and <110> direction ultimately reach a lower stress level and remain stable.

The simulated stress–strain curves of Al–4%Cu alloy single crystals under compression in different directions are shown in Figure 2b, and the crystals also exhibit strong anisotropy. In the stage of elastic deformation, three stress–strain curves show a consistent upward trend in stress, but due to different atomic arrangements in different directions, the rising heights are different. With the increase in strain, three stress–strain curves under compression all show a trend of stress decrease. Obvious differences are shown when it is compressed in different directions in the plastic deformation stage. When compressed along the <$1\bar{1}1$> direction, the stress–strain curve enters a stress-stable stage after yielding. When compressed along the <$\bar{1}12$> direction, the stress–strain curve enters a short stress plateau stage after yielding. Then, with the increase in strain, the stress started to rise, and work hardening occurred. In Fan’s study [25], it was found that the pinning of solute atoms by dislocations, the stacking and entanglement of dislocations, and a large number of immobile dislocations all contribute to the phenomenon of work hardening. When compressed along the <110> direction, both the yield strain and the strength reached their peak.

However, unlike in the case of tension, the highest slope during the elastic deformation stage does not occur along the <$1\bar{1}1$> direction but in the <110> direction. That is, when compressed along the <110> direction, the elastic modulus is at its maximum. The maximum yield strength is achieved when compressed along the <110> direction, reaching around 8 GPa, which is much larger than the yield strength under tension along that direction. When compressed in the <$\bar{1}12$> direction, the yield strain is the same as under tension conditions, both 5.9%. However, the yield strength is 5 GPa, which is greater than the yield strength in tension. When compressed along the <$1\bar{1}1$> direction, the yield strength is 4.1 GPa, which is generally the same as the yield strength at tension. This indicates that the elastic modulus is independent of its loading state in the direction of the close-packed plane of atoms, and the stress remains stable in the later stage of plastic deformation, which implies that the strengthening effect during loading counteracts the stress release caused by dislocation nucleation. When compressed along the <110> direction, the yield stress reaches 8 Gpa. In comparison with the tension state along this direction, the yield strength and elastic modulus are significantly improved. It can be found that when loaded in the two directions of <$\bar{1}12$> and <110>, the yield strength and elastic modulus are significantly related to the loading method.

### 3.2. Plastic Deformation Mechanism Analysis

In order to investigate the causes for the anisotropy of mechanical behavior of aluminum–copper alloys under tension and compression loading along different orientations, it studied the plastic deformation mechanism of the alloy under tension and compression loading along different orientations and analyzed the corresponding stress–strain curves as below.

#### 3.2.1. Tensile Deformation Mechanism Analysis

Figure 3 shows the crystal structure evolution of Al–4%Cu alloy single crystals under tension loading in different directions. Under tension along the <$\bar{1}12$> direction, if the strain reaches 5.9%, that is, when the yield strain is reached, the stress drops sharply, and HCP atoms as well as other atoms emerge in the crystal. Other atoms are defects generated under tension, mainly concentrated at the heads of HCP atoms. As the strain increases, HCP atoms increase continuously. Due to the formation of defects and HCP atoms, the stress drops sharply after reaching the yield strain under the tension along the <$1\bar{1}1$> and <$\bar{1}12$> directions. However, it shall be pointed out that the yield strain under tension in the <$1\bar{1}1$> direction is 7.9%, which is much larger than the yield strain under tension in the other two directions. Compared with the other two directions, more other atoms emerge under tension along the <$1\bar{1}1$> direction. This is because the <111> plane is a close-packed plane of atoms, which calls for greater strain to generate defects. Once defects are generated, the atomic arrangement around the defects will be disrupted, resulting in more other atoms. Figure 3c shows the evolution of crystal structure under tension along the <110> direction, and it is found that periodic arrangement of HCP atomic planes occurs in the process of tension, resulting in fewer other atoms and fewer defects. Therefore, under tension along the <110> direction, although the stress decreases in the plastic deformation stage, there is no surging decrease in stress like in the other two directions. That is because defects would not be generated easily under tension along the <110> direction.

Figure 4 shows the changes in dislocation density of the Al–4%Cu alloy single crystal under tension in different directions. It has been found that almost no Shockley partial dislocations would be generated during the elastic deformation stage under tension in any direction. However, the dislocation density increases sharply after reaching a certain strain. In accordance with Table 1, it can be found that the strain point where dislocation proliferation occurs shows yield strain. The reasons are as follows: with the emergence of dislocations, the integrity of the crystal structure is disrupted and the atomic binding forces decrease, which releases the loading stress of the crystal. This is consistent with dislocation theory.

Figure 5 shows the dislocation structure diagram (with a strain of 14%) of Al–4%Cu alloy single crystals under tension along different directions. As shown by the analysis on dislocations, defects generated over the tension loading process of the crystal in Figure 3 are dislocations, and the HCP atoms are stacking faults. The crystal features a relatively high dislocation density under tension along atomic binding forces, mainly consisting of Shockley partial dislocations and Stair-rod dislocations. The motion of Shockley partial dislocations leads to the HCP atomic stacking faults [26], while the Stair-rod dislocation is prone to form Lomer–Cottrell dislocation locks [27,28,29]. The formation process is shown in Figure 6, and the formation of this dislocation lock could strengthen the material. Therefore, it can be observed that in Figure 2a, the flow stress in the later stage of plastic deformation under tension along the <$1\bar{1}1$> direction is higher than that in the other two directions. Under tension along the <$\bar{1}12$> direction and <110> direction, it is dominated by Shockley partial dislocations, so the flow stress is not high in the later stage of plastic deformation. It shall be pointed out that the dislocation density is lower under tension along the <110> direction, and it is dominated by the HCP atomic stacking faults caused by Shockley partial dislocations. These stacking faults are intrinsic stacking faults. Due to the loose atomic arrangement of the <110> plane itself, it does not call for a large number of dislocation movements for atomic dislocation when subjected to tensile stress, so stress release is carried out in the form of stacking faults. The periodic arrangement of stacking faults shown in Figure 3c could strengthen the material [30], which is also the main reason for the stress increase in the later stage of plastic deformation under tension along the <110> direction as shown in Figure 2a.

The formation of various defects in metal alloys can cause changes in the atomic lattice, and changes in crystal potential energy can be used to describe changes in the atomic lattice. Therefore, it could observe changes in crystal structure more intuitively by calculating the atomic potential energy during the deformation process of the crystal model. Figure 7 shows the crystal potential energy analysis of Al–4%Cu alloy single crystals under tension in different directions. Under tension along the <$\bar{1}12$> direction, when the strain is large enough, high-energy atoms appear. These high-energy atoms are caused by atomic dislocation displacement, also known as stacking faults, due to dislocation glide. As the motion of dislocations produces sessile dislocation, the energy is highest at the intersection of stacking faults, resulting in the work hardening phenomenon [25,31,32]. Therefore, in Figure 2a, as the strain increases, the stress starts to rise during the later stage of plastic deformation under tension in this direction. Under tension in the <$1\bar{1}1$> direction, with the increase in strain, stress concentration occurs at the stacking faults and other atoms, which may be caused by dislocation tangles. However, due to the tension on the close-packed plane, a large number of dislocations are generated, which ensures outstanding plastic performance. Such also serves as the reason for the absence of work hardening under tension in this direction. Under tension in the <110> direction, it is found that the high-energy atoms mainly exist at the dislocation of HCP atoms, in light of Figure 3c. It can be seen intuitively that the stacking fault presents a periodic distribution. A “box” is developed inside the crystal to constrain it and thus play the reinforcing effect. It is this “box” that hinders the formation of dislocations. Therefore, it is observed in Figure 5 that the dislocation density is lower under tension along the <110> direction.

#### 3.2.2. Compressive Deformation Mechanism Analysis

In the previous section, the analysis of the stress–strain curve revealed that the loading method also has a significant impact on the mechanical behavior of aluminum–copper alloys. Therefore, in the following part, this paper will study the deformation mechanism during compression loading and compare it with the performance over tension.

Figure 8 shows the evolution of crystal structure during compression in three different directions. In the elastic deformation stage, the crystal maintains a single FCC structure. As the strain increases, the crystal enters the plastic deformation stage, where the main plastic deformation mechanisms are dislocations and stacking faults. When compressed in the <$\bar{1}12$> direction, if the strain reaches 5.9% of the yield strain, the increase in strain leads to lattice failure, resulting in defects. The formation and movement of defects cause a decrease in stress in the plastic deformation stage while generating HCP atoms, which is consistent with the crystal structure change under tension in that direction. The structural change of the crystal during compression along the <$1\bar{1}1$> direction is similar to the structural change process under tension. As the strain increases, defects and HCP atoms are formed. In comparison with tension conditions, crystal structure during compression along the <110> direction does not show periodic HCP atomic planes alike but is still dominated by defects and HCP atoms as the strain increases.

Figure 9 shows the variation of dislocation density under compression in different directions. In accordance with observations, regardless of the direction of compression, the dislocation density increases with strain. It is mainly dominated by Shockley partial dislocations, with a small amount of Stair-rod dislocations. That also serves as the reason for the generally consistent flow stress in the three compression directions in the later stage of plastic deformation.

Figure 10 shows the dislocation structure of Al–4%Cu alloy single crystals in different directions (with a strain of 14%). In contrast with tension, the plastic deformation process under compression in different directions is mainly dominated by Shockley partial dislocations and Stair-rod dislocations, exhibiting similar deformation mechanisms under compression in different directions. It shall be pointed out that a large number of dislocations are generated over the compression loading along the <110> direction. The dislocation density is relatively high, much higher than that under tension. That should be attributed to the facts below. As the atomic arrangement of the <110> plane itself was not dense originally, compressed loading would cause an increasingly dense arrangement. In such a case, the stress could only be released through dislocations.

Figure 11 is the crystal potential energy analysis of Al–4%Cu alloy single crystals under compression in different directions. This figure shows that in the process of compression along different directions, high-potential atoms exist at the dislocation and dislocation entanglement sites, no matter what the compression direction is. This also indicates that compressive loading causes the crystal to undergo the same plastic deformation mechanism, which is independent of the compression direction.

#### 3.2.3. Analysis of the Mechanism of the Loading Methods’ Impact on Mechanical Properties

To sum up, some interesting phenomena can be observed. Firstly, different loading methods along the <$1\bar{1}1$> direction do not affect the mechanical properties of the crystal. The yield strength under tension and compression loading is roughly the same. This is because the <111> plane is the close-packed plane of FCC-structured crystals, which is the most stable stacking method. When subjected to external forces, the deformation of the crystal remains the same, mainly dominated by dislocations and stacking faults. Secondly, the loading direction of the highest yield strength varies with different loading methods. When loaded in the <$1\bar{1}1$> under tension, the yield strength is highest, and when loaded in the <110> under compression, the yield strength is highest. In accordance with the observation on the dislocation density, the dislocation density in the two corresponding loading directions under tension and compression is the highest, indicating that the high dislocation density has a strengthening effect on material strength, which is also in line with the dislocation enhancement theory.

## 4. Conclusions

This article studies the anisotropy of aluminum–copper alloys with molecular dynamics methods. Three different orientations of Al–4%Cu alloy crystals were selected for tension and compression loading, and the mechanical properties and plastic deformation mechanisms of the results were analyzed. The main conclusions are as follows:

(1)The Al–4%Cu alloy exhibits significant anisotropy under loading in different directions. The elastic modulus and yield strength are the highest under tension along the <$1\bar{1}1$> direction. The trend is completely opposite under compression. Al–4%Cu alloy also shows significant differences under different loading conditions. However, the <$1\bar{1}1$> direction is not sensitive to the loading condition, and the yield strength is basically the same under tension and compression.(2)Changes in atomic structure and atomic potential energy of Al–4%Cu alloy crystals under loading in different directions indicate that crystals under loading in different directions have a single FCC structure in the elastic stage and dislocation glide generates stacking faults in the plastic deformation stage. High-potential energy atoms are mainly concentrated at the stacking faults and dislocation entanglements.(3)Under tension along the <$1\bar{1}1$> direction, the close-packed plane of atoms leads to large atomic binding forces, which results in a high yield strength. In addition, the Stair-rod dislocation forms a Lomer–Cottrell dislocation lock, which has a strengthening effect on the material. Periodic arrangement of HCP atomic planes appears under tension along the <110> direction. That is, the deformation mechanism is dominated by stacking faults. That is because the loose atomic arrangement of the <110> plane could easily lead to atomic misalignment.(4)When compressed in different directions, the plastic deformation mechanism is mainly dominated by dislocations and stacking faults. When compressed along the <110> direction, it has a relatively high dislocation density. That should be attributed to the facts below. As the atomic arrangement of the <110> plane itself was not dense originally, compression loading would cause an increasingly denser arrangement. In such a case, the stress could only be released through dislocations. Therefore, the yield strength along <110> is the largest.(5)Changes in dislocation density are closely related to the microstructure and stress of the alloy. In the case of dislocation multiplication, as dislocation density increases, stress decreases. When dislocation density stabilizes, stress increases, as there is no new dislocation nucleation. During the tension or compression loading process, the higher the dislocation density along the loading direction, the larger the rheological stress in the plastic deformation stage in that loading direction, which is consistent with the dislocation strengthening theory.

## Figures and Tables

**Figure 1 materials-17-04162-f001:**
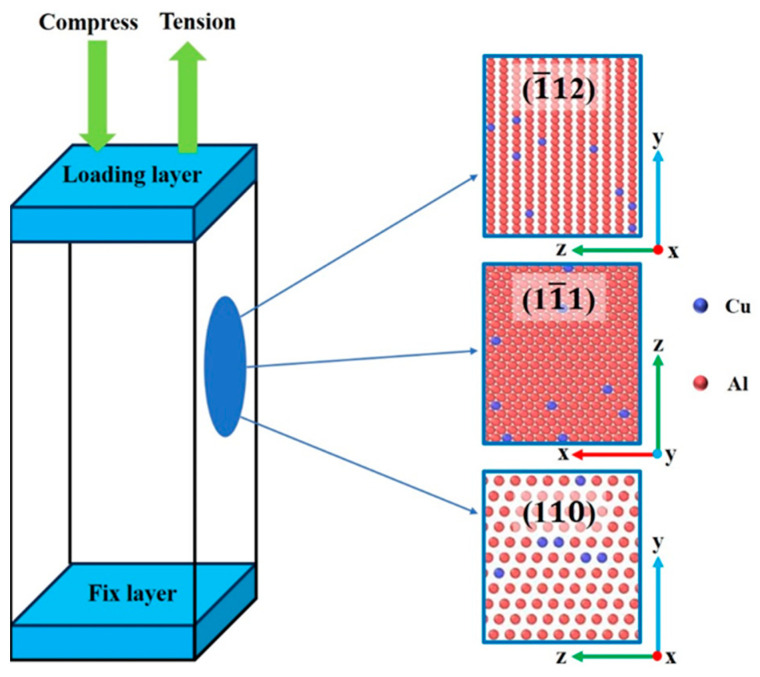
Al–4%Cu alloy single crystal model loading schematic diagram.

**Figure 2 materials-17-04162-f002:**
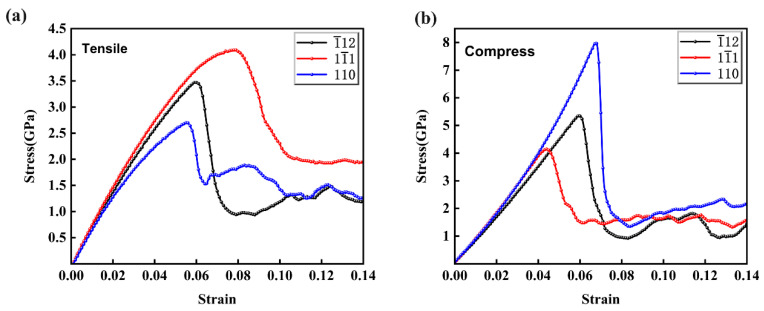
Simulated stress–strain curves under different loading directions. (**a**) tensile; (**b**) compression.

**Figure 3 materials-17-04162-f003:**
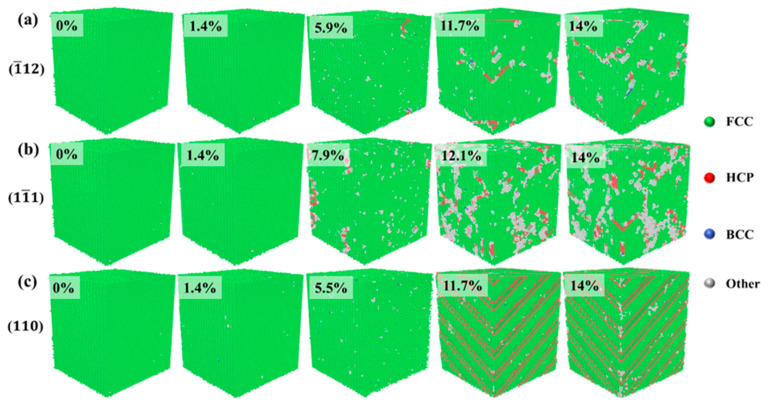
Evolution diagram of crystal structure under tension in different directions (the percentages represent different strains).

**Figure 4 materials-17-04162-f004:**
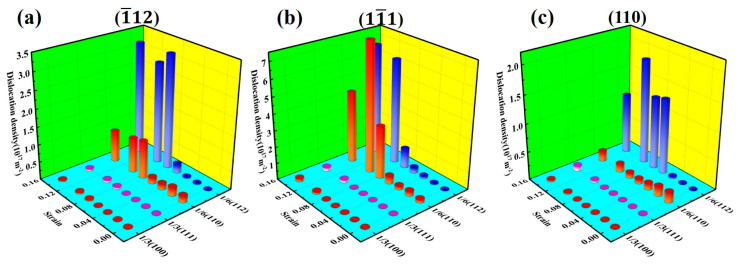
Changes in dislocation density of different dislocation under tension in different directions.

**Figure 5 materials-17-04162-f005:**
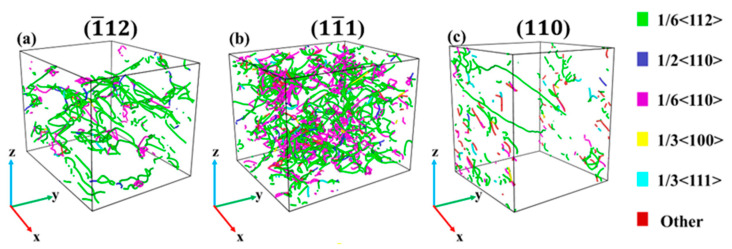
Dislocation structure diagram under tension in different directions (with a strain of 14%).

**Figure 6 materials-17-04162-f006:**
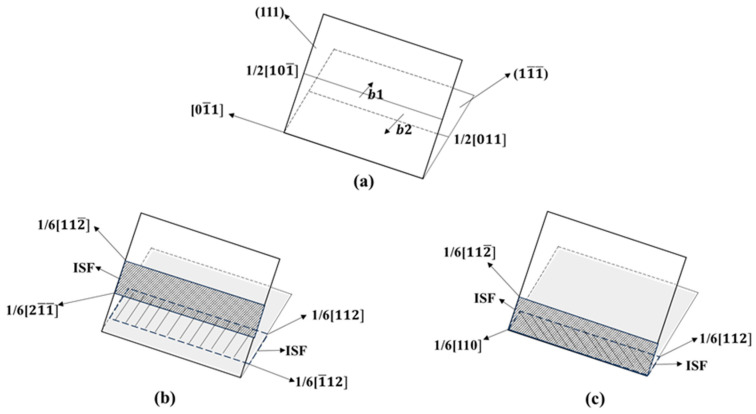
Formation process of Lomer–Cottrell dislocation lock. ((**a**) two dislocations in different directions; (**b**) two dislocations generate two stacking faults; (**c**) two stacking faults intersect to form Lomer-Cottrell dislocation lock).

**Figure 7 materials-17-04162-f007:**
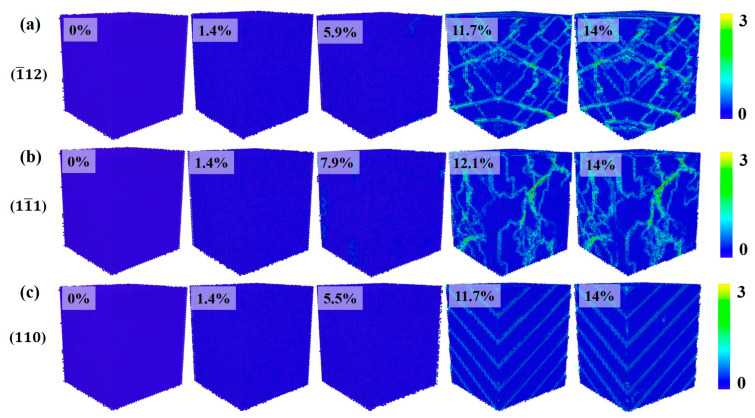
Analysis of crystal potential energy under tension in different directions (the percentages represent different strains).

**Figure 8 materials-17-04162-f008:**
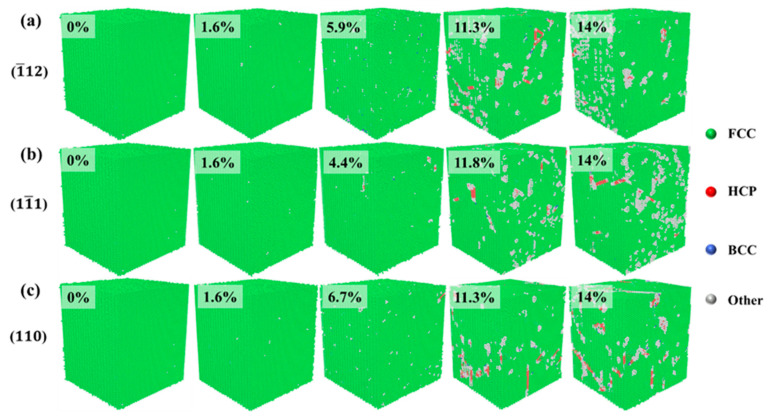
Evolution diagram of crystal structure under compression in different directions (the percentages represent different strains).

**Figure 9 materials-17-04162-f009:**
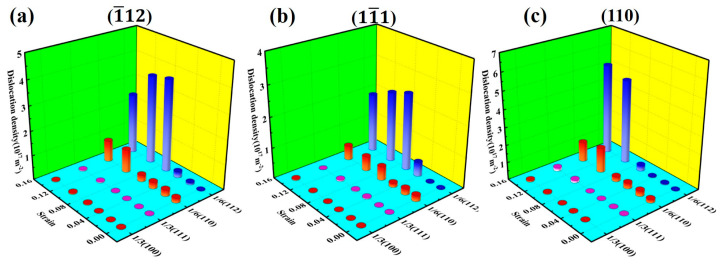
Changes in dislocation density of different dislocation under compression in different directions.

**Figure 10 materials-17-04162-f010:**
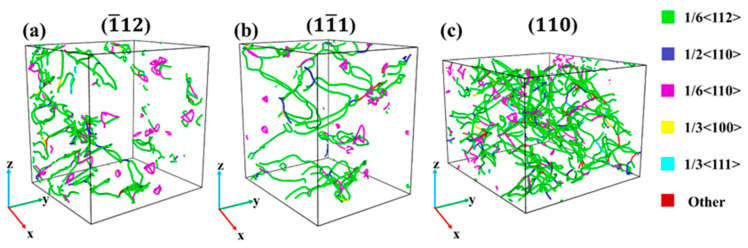
Dislocation structure diagram under compression in different directions (with a strain of 14%).

**Figure 11 materials-17-04162-f011:**
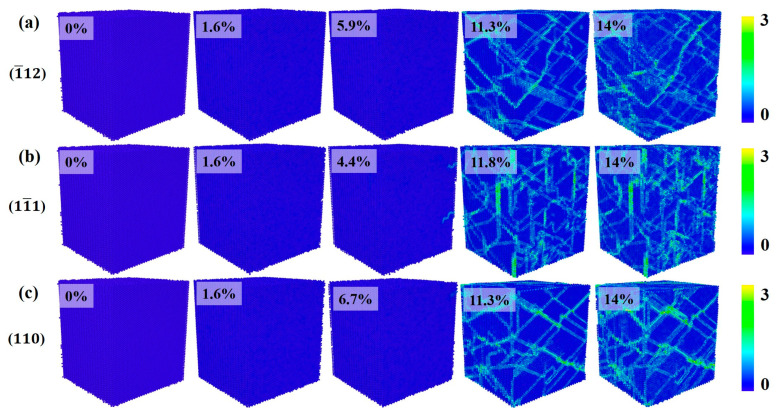
Analysis diagram of crystal potential energy under compression in different directions (the percentages represent different strains).

**Table 1 materials-17-04162-t001:** Yield strength and yield strain under tension and compression loading in different directions.

Loading Direction and Method	Yield Strength (GPa)	Yield Strain (%)
<$\bar{1}12$> (Tension)	3.45	5.90
<$1\bar{1}1$> (Tension)	4.15	7.90
<110> (Tension)	2.60	5.50
<$\bar{1}12$> (Compression)	5.00	5.90
<$1\bar{1}1$> (Compression)	4.10	4.40
<110> (Compression)	8.00	6.70

## Data Availability

The original contributions presented in the study are included in the article, further inquiries can be directed to the corresponding author.

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
