# Peer review of "Molecular Dynamics Simulation Study of Aluminum–Copper Alloys’ Anisotropy under Different Loading Conditions and Different Crystal Orientations"

_materials, 2024, doi:10.3390/ma17164162_

Round 1

Reviewer 1 Report

Comments and Suggestions for Authors

Dear Authors,

Your paper is iteresting but I was mislead. Normally the simulatoin results ar ewell presented separately from laboratory results that are used for calibrating or, better, prove the good quality of the model they develop. 

On this base, please find a list of comments/suggestions:

The title is misleading: the Authors have simulated the Al-Cu anisotropy under different loading condition and not proved it by mean of experimental laboratory activity. Please revise the Title introducing that all the activity is “a modelling” without any experimental confirmation.

Line 30-31: it is suggested a less dramatic sentence: “ This research aims to provide a reference… , instead of your: “The research results provide important references….”

Line 112: The NPT meaning should be introduce before using it.

Line 115: same for NVT

Line 18. Report the house/developer of LAMMPS and further details as for guidelines, idem for CAN, OVITO and DXA.

Line 123-124: the stress-strain curves are experimental or simulated? In the case of simulated, please state it clearly.

Caption of Fig.2: please, explain if this curves are simulated or experimental.  As a general comment, please help the reader stating that all the tests are simulated. Same approach at line 136: Fig 28a) does not show the stress-strain curves, but the simulation model of the stress-strain curves obtained by the authors.

As a more general comment: the Authors have opened an important window on a particular behaviour and have tried to enter in the real atomic- deformation mechanism acting in a Al-Cu alloy.

he only problem is that they have not yet give an experimental confirmation of their modelled behaviour.

The results are reported in a way that is confusing: as they were the real experimental case. It is suggested to the Authors to state clearly that this is an attempt to understand and describe from a microscopic perspective. Moreover that, as a preliminary step, all the conclusion and simulation results should be proven by mean of targeted experimental activity.

So, this paper could represent a preliminary step for an experimental future laboratory activity:

1)      Single crystal (SC)  manufacturing

2)      SC tensile /compressive testing

3)      Comparison of experimental with your simulated results .

Author Response

Dear Editors and Reviewers.

Thank you for your letter and for the reviewers' comments concerning our manuscript entitled " Molecular Dynamics Simulation Study of Aluminum Copper Alloys’ Anisotropy under Different Loading Conditions and Different Crystal Orientations”. Those comments are all valuable and very helpful for revising and improving our paper, as well as the important guiding significance to our researches. We have studied comments carefully and have made correction with red mark in the revised manuscript which we hope meet with approval. We would like to show the details as follows:

Reviewer 1:

  1. The title is misleading: the Authors have simulated the Al-Cu anisotropy under different loading condition and not proved it by mean of experimental laboratory activity. Please revise the Title introducing that all the activity is “a modelling” without any experimental confirmation.

Response: Thank you for pointing this out. The title has been corrected in the revised manuscript.

  1. Line 30-31: it is suggested a less dramatic sentence: “ This research aims to provide a reference… , instead of your: “The research results provide important references….”

Response: We sincerely thank the reviewer for careful reading. As suggested by the reviewer, we have corrected this sentence in the revised manuscript.

Line 112: The NPT meaning should be introduce before using it;Line 115: same for NVT; Line 18. Report the house/developer of LAMMPS and further details as for guidelines, idem for CNA, OVITO and DXA.

Response: We sincerely appreciate the valuable comments. We have added the meaning and more detail in the revised manuscript.

  1. Line 123-124: the stress-strain curves are experimental or simulated? In the case of simulated, please state it clearly.

Response: We feel sorry for our carelessness. The stress-strain curves are simulated. We have corrected it in the revised manuscript.

  1. Caption of Fig.2: please, explain if these curves are simulated or experimental.  As a general comment, please help the reader stating that all the tests are simulated. Same approach at line 136: Fig 28a) does not show the stress-strain curves, but the simulation model of the stress-strain curves obtained by the authors.

Response: We feel sorry for our carelessness. The stress-strain curves are simulated. We have thoroughly checked the relevant carelessness throughout the manuscript and corrected them in the revised manuscript.

As a more general comment: the Authors have opened an important window on a particular behaviour and have tried to enter in the real atomic- deformation mechanism acting in a Al-Cu alloy.he only problem is that they have not yet give an experimental confirmation of their modelled behaviour.The results are reported in a way that is confusing: as they were the real experimental case. It is suggested to the Authors to state clearly that this is an attempt to understand and describe from a microscopic perspective. Moreover that, as a preliminary step, all the conclusion and simulation results should be proven by mean of targeted experimental activity.So, this paper could represent a preliminary step for an experimental future laboratory activity:

1)      Single crystal (SC)  manufacturing

2)      SC tensile /compressive testing

3)    Comparison of experimental with your simulated results .

Response: We sincerely appreciate the valuable comments. At present, we have designed targeted experiments and are still in the experimental stage. We look forward to proving the conclusions mentioned in this article in the future.

Reviewer 2 Report

Comments and Suggestions for Authors

In the manuscript the authors conducted the loading simulation on Al-4%Cu alloy crystals at the microscopic scale with the molecular dynamics technology. The embedded atom method developed by Liu et al. was used in the simulation proces. A single crystal model of an aluminum-copper alloy was established, and tensile and compressive loading of the alloy was simulated on a microscopic scale.

The topic of the publication may be of interest to the scientific community involved in optimizing processing technology and preparation methods of aluminum copper alloys. 

The manuscript is well organized. The work is sufficiently detailed and conclusive. The findings are properly discussed and explained to the reader. Conclusions reflect the research results obtained and are consistent with the arguments presented. 

The references contain 31 items, most from the last two years. The references are appropriate. There are no inappropriate self-citations by the authors.

Given the merits indicated above, in my opinion the work is worth publishing. 

The manuscript requires only minor revision to guarantee publication in Materials: 

- The titles of figures 3, 7 and 8 lack information on what the values expressed in percentages represent.

Author Response

Dear Editors and Reviewers.

Thank you for your letter and for the reviewers' comments concerning our manuscript entitled " Molecular Dynamics Simulation Study of Aluminum Copper Alloys’ Anisotropy under Different Loading Conditions and Different Crystal Orientations”. Those comments are all valuable and very helpful for revising and improving our paper, as well as the important guiding significance to our researches. We have studied comments carefully and have made correction with red mark in the revised manuscript which we hope meet with approval. We would like to show the details as follows:

Reviewer 2

The manuscript requires only minor revision to guarantee publication in Materials: 

  1. The titles of figures 3, 7 and 8 lack information on what the values expressed in percentages represent.

Response: Thank you for your nice comments on manuscript. The percentages in Figures 3, 7, and 8 represent different strains. We have added the description in the revised manuscript.

Reviewer 3 Report

Comments and Suggestions for Authors

This paper conducted the loading simulation on Al-4%Cu alloy crystals at the microscopic scale with the molecular dynamics technology. The research results provide references for optimizing the processing technology and preparation methods of aluminum copper alloy materials.

The article is rather well written and contains useful data.

Some minor changes suggested are:

-          The figures design needs revision in terms of clarity

-          Remove background from Figure 2.

-          Insert Table in the text and not Tab

-          Add a comparison with literature in Results Discussion section

-          Conclusion section needs more insights and a phrase outlining perspectives.

Comments on the Quality of English Language

English is OK. Minor changes in terms of grammar are needed.

Author Response

Dear Editors and Reviewers.

Thank you for your letter and for the reviewers' comments concerning our manuscript entitled " Molecular Dynamics Simulation Study of Aluminum Copper Alloys’ Anisotropy under Different Loading Conditions and Different Crystal Orientations”. Those comments are all valuable and very helpful for revising and improving our paper, as well as the important guiding significance to our researches. We have studied comments carefully and have made correction with red mark in the revised manuscript which we hope meet with approval. We would like to show the details as follows:

Reviewer 3

This paper conducted the loading simulation on Al-4%Cu alloy crystals at the microscopic scale with the molecular dynamics technology. The research results provide references for optimizing the processing technology and preparation methods of aluminum copper alloy materials.

The article is rather well written and contains useful data. Some minor changes suggested are: 

  1. The figures design needs revision in terms of clarity.

Response: We feel sorry for our carelessness. We have checked the figures quality throughout the manuscript, and corrected them in the revised manuscript.

  1. Remove background from Figure 2.

Response: We feel sorry for our carelessness. We have corrected Figure 2 in the revised manuscript.

  1. Insert Table in the text and not Tab.

Response: Thank you for pointing this out. We have corrected it in the revised manuscript.

  1. Add a comparison with literature in Results Discussion section.

Response: We sincerely appreciate the valuable comments. We have added comparison with literature in the revised manuscript.

  1. Conclusion section needs more insights and a phrase outlining perspectives.

Response: Thanks for your suggestion. In the revised manuscript, some conclusions have been simplified and modified.

Round 2

Reviewer 1 Report

Comments and Suggestions for Authors

One remark: please, take care about this comment. The modification has not been implemented.

1.        Line 30-31: it is suggested a less dramatic sentence: “ This research aims to provide a reference… , instead of your: “The research results provide important references….”

Response: We sincerely thank the reviewer for careful reading. As suggested by the reviewer, we have corrected this sentence in the revised manuscript.

Author Response

please, take care about this comment. The modification has not been implemented.

Response: We feel very sorry for our carelessness. We have modified this sentence in the revised manusript.
